# Unraveling the Genetic Control of Pigment Accumulation in Physalis Fruits

**DOI:** 10.3390/ijms25189852

**Published:** 2024-09-12

**Authors:** Wennan Zhao, Haiyan Wu, Xiaohan Gao, Huimei Cai, Jiahui Zhang, Chunbo Zhao, Weishu Chen, Hongyu Qiao, Jingying Zhang

**Affiliations:** Modern Vegetable Industry Technology and Germplasm Resource Innovation Team, Northeast Asia Special Germplasm Resource Conservation and Innovation Center Vegetable Breeding Technology Innovation Team, College of Horticulture, Jilin Agricultural University, Changchun 130118, China

**Keywords:** Physalis, fruit coloration, carotenoid biosynthesis, gene regulation, transcription factors

## Abstract

*Physalis pubescens* and *Physalis alkekengi*, members of the Physalis genus, are valued for their delicious and medicinal fruits as well as their different ripened fruit colors—golden for *P. pubescens* and scarlet for *P. alkekengi*. This study aimed to elucidate the pigment composition and genetic mechanisms during fruit maturation in these species. Fruit samples were collected at four development stages, analyzed using spectrophotometry and high-performance liquid chromatography (HPLC), and complemented with transcriptome sequencing to assess gene expression related to pigment biosynthesis. β-carotene was identified as the dominant pigment in *P. pubescens*, contrasting with *P. alkekengi*, which contained both lycopene and β-carotene. The carotenoid biosynthesis pathway was central to fruit pigmentation in both species. Key genes *pf02G043370* and *pf06G178980* in *P. pubescens*, and *TRINITY_DN20150_c1_g3*, *TRINITY_DN10183_c0_g1*, and *TRINITY_DN23805_c0_g3* in *P. alkekengi* were associated with carotenoid production. Notably, the MYB-related and bHLH transcription factors (TFs) regulated zeta-carotene isomerase and β-hydroxylase activities in *P. pubescens* with the MYB-related TF showing dual regulatory roles. In *P. alkekengi*, six TF families—bHLH, HSF, WRKY, M-type MADS, AP2, and NAC—were implicated in controlling carotenoid synthesis enzymes. Our findings highlight the intricate regulatory network governing pigmentation and provide insights into Physalis germplasm’s genetic improvement and conservation.

## 1. Introduction

*Physalis pubescens* and *Physalis alkekengi* are two well-known herbaceous plants of the Physalis genus within the Solanaceae family, which is widely cultivated in China [1]. *P. alkekengi*, also known as Chinese lantern, is a rhizomatous perennial herb with red–orange fruits and fruiting calyces, and it is the only native Eurasian species of the genus *Physalis*. It has long been grown throughout China, Japan, and Europe, and it has a long history in the therapy of cough, excessive phlegm, pharyngitis, sore throat, dysuria, pemphigus, eczema, and jaundice in traditional Chinese medicine [2]. In contrast, *P. pubescens* is a weedy annual herb native to the Americas with yellow fruits and calyces [3], which is commonly used in traditional Chinese medicine due to its anti-inflammatory, antipyretic, diuretic, antitussive, and antitumor effects [4]. Currently, they are extensively cultivated in Jilin Province, Heilongjiang Province, and the Inner Mongolia Autonomous Region [5]. The berry fruits of *P*. *pubescens* and *P*. *alkekengi* are prized for their palatable tastes and are also valued for their ornamental appeal and medicinal attributes [6,7]. The fully mature fruits of *P*. *pubescens*, commonly known as “golden tomatillo”, have a vivid golden color, whereas those of *P. alkekengi*, referred to as “scarlet tomatillo”, exhibit a deep orange–red hue when mature [8].

The color of the fruit is a significant economic trait in plant breeding with specific types and varying concentrations of pigments determining the coloration and visual spectrum of plant organs. Natural pigments in plants can be categorized into four main groups: chlorophylls, carotenoids, flavonoids, and betalains [9,10]. Chlorophylls, found primarily in the chloroplasts, include chlorophyll a and chlorophyll b. Carotenoids, also presented in the chloroplasts, consist of various pigments, including orange carotene, yellow xanthophylls, and red lycopene. Different combinations of these pigments produce yellow, orange, and other colors in plants [11]. Flavonoids primarily comprise anthocyanins, aurones, and chalcones [12]. Betalains, a class of alkaloids, display a range of hues in plant tissues, such as purple, white, and red. There are two main types of classification for betalains: betaxanthins, which are yellow, and betacyanins, which are red [13,14]. During the early stages of fruit development, the predominant green hue is primarily attributed to the abundant presence of chlorophylls. The shift in fruit coloration results from the gradual breakdown of chlorophylls in the cells of the fruit along with the concurrent synthesis, buildup, and exposure of pigments like carotenoids. As a result, the fruit changes color from an initial green hue to its mature coloration [15].

In the plant carotenoid biosynthetic pathway, lycopene cyclase enzymes, which comprise lycopene ε-cyclase (*LCYE*) and lycopene β-cyclase (*LCYB*), are pivotal for converting lycopene into α-carotene and β-carotene, marking a significant branch point in carotenogenesis. α-Carotene is then converted to lutein under the catalysis of carotenoid β-ring hydroxylase and carotenoid ε-hydroxylase [16]. β-Carotene is further processed through the action of β-carotene hydroxylase (*BCH*), which converts it to β-cryptoxanthin and subsequently to zeaxanthin. The synthesis of capsanthin, a red ketocarotenoid responsible for imparting a vivid red hue to fruits, commences with the epoxidation of zeaxanthin by a specific epoxidase enzyme to yield epoxyzeaxanthin. Following this, capsanthin–capsorubin synthase (*CCS*) triggers the cyclization of epoxyzeaxanthin, producing capsanthin [17].

The variations in carotenoid contents are a crucial factor in determining the color of fruits and are tightly controlled by the activity of genes involved in the carotenoid biosynthesis pathway. For example, differences in carotenoid accumulation among pepper fruits are primarily due to variations in the gene expression levels and enzymatic activities of key carotenoid biosynthetic enzymes, such as *LCYB*, *LCYE*, and *CCS*. These findings have shed light on the complex regulatory mechanisms governing the biosynthesis of both carotenoids and flavonoids in fruits [18]. Additionally, in tomatoes, levels of phytoene desaturase (PDS) and zeaxanthin epoxidase (ZDS) are positively correlated with the overall carotenoid content, particularly lycopene. These correlations underscore the pivotal role that specific genes play in the carotenoid biosynthetic pathway and their regulatory impact on the fruit’s pigment composition [19,20].

Previous studies have indicated that the primary pigments in *P. pubescens* and *P. alkekengi* fruits are carotenoids [21]. However, the metabolic mechanisms and regulatory genes that influence the color changes during fruit development have been scarcely reported to date. Therefore, our study aims to investigate the pigment metabolism pathways, specifically the carotenoid biosynthesis pathway (map00906), during the development of Physalis fruit using High-Performance Liquid Chromatography (HPLC) and transcriptome sequencing. By doing so, we aim to elucidate the metabolic characteristics of the pigments in Physalis fruit and identify key genes and transcription factors that play a crucial role in the coloration during the development phase. This investigation successfully quantified the pigment traits of the two types of Physalis fruits, identified crucial genes and transcription factors implicated in pigment formation during fruit maturation, and contributed to the development of fruit pigment resources and Physalis germplasm conservation and genetic enhancement.

## 2. Results

### 2.1. Fruit Pigment Contents Comparisons

#### 2.1.1. Total Flavonoids Analysis

The information in Table 1 demonstrates the overall flavonoid contents identified in the fruits of *P. pubescens* (Y) and *P. alkekengi* (R) across four distinct developmental stages: green ripening stage S1, color conversion stage S2, firm ripening stage S3, and full ripening stage S4. In *P. pubescens* (Y) fruits, the total flavonoid contents varied from 112.64 to 341.65 μg/g, with the highest concentration recorded at the full ripening stage (YS4) at 341.65 μg/g. On the other hand, the total flavonoid contents in *P. alkekengi* (R) fruits peaked at the full ripening stage (RS4) with a value of 339.36 μg/g. The order of flavonoid contents ranking, from highest to lowest, was consistent for both *P. pubescens* and *P. alkekengi* across different ripening stages: full ripening stage > firm ripening stage > color conversion stage > green ripening stage. Notably, during the later phases of fruit development in both *P. pubescens* and *P. alkekengi*, a sudden surge in total flavonoid contents may be due to external stress factors, leading to significantly heightened levels compared to earlier developmental stages. Nevertheless, there were no significant differences in the overall flavonoid contents between the firm and full ripening stages in both *P. pubescens* and *P. alkekengi*. These indicated that the alteration in flavonoid contents between these stages may not substantially impact the fruit colors. Except for the green ripening stage (S1), the content of flavonoids in *P. pubescens* is higher than that in *P. alkekengi* during the other three developmental phases, which may contribute to the more pronounced yellow coloration observed in the fruits of *P. pubescens* (Figure 1).

#### 2.1.2. Total Anthocyanin Analysis

The cumulative anthocyanin contents of the fruits belonging to *P. pubescens* (Y) and *P. alkekengi* (R) at four distinct developmental phases (S1–S4) are detailed in Table 2. At the commencement of fruit development (S1), *P. alkekengi* (R) exhibited a higher anthocyanin content in comparison to *P. pubescens* (Y), and both fruits demonstrated a progressive increase in total content during their growth. Analysis revealed that the total anthocyanin contents in *P. pubescens* fruits varied from 11.03 to 32.75 nmol/g, with the highest point at the firm ripening stage (YS3) and full maturity stage (YS4), at 30.83 nmol/g and 32.75 nmol/g, respectively. In *P. alkekengi* fruits, the overall anthocyanin levels varied between 25.98 and 58.73 nmol/g throughout the four phases with the peak measurement seen at the complete ripening stage (RS4), hitting 58.73 nmol/g. Throughout the four developmental periods, the content of anthocyanins in *P. alkekengi* consistently exceeds that of *P. pubescens*. This elevated level of anthocyanins may account for the more intense red color observed in *P. alkekengi* fruits as compared to *P. pubescens* (Figure 1). Anthocyanins accumulated substantially within the fruits of *P. pubescens* and *P. alkekengi* during the firm and full ripening stages. As a crucial pigment constituent influencing fruit coloration, anthocyanins may play a significant role in the coloring process of the fruit during the later developmental phases.

#### 2.1.3. Carotenoid and Chlorophyll Content Analysis

The levels of carotenoids in the fruits of *P. pubescens* (Y) and *P. alkekengi* (R) during the four developmental phases (S1–S4) have been detailed in Table 3. Both *P. pubescens* and *P. alkekengi* fruits displayed increased carotenoid contents as they progressed through the developmental stages. The carotenoid contents in *P. pubescens* (Y) fruits ranged from 72.73 to 360.07 μg/g, with the highest concentration observed at the full ripening stage (YS4), reaching 360.07 μg/g, significantly exceeding levels at other stages. Similarly, the carotenoid contents in *P. alkekengi* fruits ranged from 175.07 to 506.81 μg/g across the four developmental stages, with the peak concentration at the full ripening stage (RS4) amounting to 506.81 μg/g, which was notably higher than levels in the early stages of fruit development (green ripening and color conversion stages). Except for the firm ripening stage (S3), *P. alkekengi* exhibits higher carotenoid content compared to *P. pubescens* across other developmental phases, which may contribute to the differential coloration observed in their fruits (Figure 1). As carotenoids are the primary pigment compounds that determine fruit color, the high accumulation of carotenoids in the final stage of fruit maturity in *P. pubescens* and *P. alkekengi* indicated that carotenoids were the main factors determining the final color of those fruits.

As demonstrated in Table 4, the levels of chlorophyll in *P. pubescens* exhibited a range of 134.33 to 561.89 μg/g, while in *P. alkekengi*, the range was observed to be 34.96 to 427.87 μg/g. Upon examination of the growth stages of *P. pubescens* and *P. alkekengi*, it was observed that during the early phases of fruit development, the chlorophyll contents were two to five times greater than those of carotenoids (Figure 2). As the fruits of both species matured, there was a gradual decline in chlorophyll levels and a reduction in their proportion to carotenoids, while the presence of carotenoids increased. This demonstrated a negative correlation between the levels of chlorophyll and carotenoids. This phenomenon elucidated the initial green hue of the fruits where the abundance of chlorophyll masked the presence of carotenoids. As the fruits ripened, the rate of chlorophyll degradation exceeded its synthesis, leading to an increase in carotenoid accumulation and shifts in fruit coloration.

#### 2.1.4. Comparative Analysis of Lutein, β-Carotene, and Lycopene Contents in *P. pubescens* and *P. alkekengi* Across Different Fruit Growth Stages

The technique of HPLC was employed for the precise measurements of lutein, β-carotene, and lycopene levels in the fruit samples of *P. pubescens* (Y) and *P. alkekengi* (R) at various stages of development (S1–S4). The outcomes are detailed in Table 5 and Table 6.

In *P. pubescens* (Y), lutein contents demonstrated a decline across the four developmental stages, peaking at 3.2614 μg/g during the green ripening stage (YS1) and dropping to 0.0172 μg/g at the firm ripening stage (YS3). β-carotene levels displayed variability from 0.0320 to 10.9035 μg/g, peaking at the full ripening stage (YS4) and dropping to the lowest at the color conversion stage (YS2). Lycopene contents were highest during the green ripening stage, reaching 21.8701 μg/g, significantly surpassing levels in other stages. Despite lutein levels being the most abundant during the green ripening stage, the fruits did not exhibit a distinct yellow hue, which was potentially due to the masking effects of high chlorophyll levels. β-carotene contents notably increased from the color conversion stage to the firm ripening stage, indicating its considerable impact on fruit coloration. Lycopene contents gradually decreased with fruit maturation, suggesting it may not be a primary pigment influencing the color of *P. pubescens* fruits.

For *P. alkekengi* (R), lutein levels ranged from 0.4724 to 2.7591 μg/g. The β-carotene contents in *P. alkekengi* exhibited the most significant increase, reaching a peak of 36.9993 μg/g at the full ripening stage (RS4). This value was notably 310 times higher than the content of 0.1190 μg/g observed at the green ripening stage. Lycopene contents ranged from 5.6711 to 63.4487 μg/g with the highest levels recorded at the color conversion stage and the lowest at the green ripening stage. Except for the green ripening stage, the content of lycopene in the fruit of *P. alkekengi* is significantly higher than that in *P. pubescens* during other developmental stages, which may be the main reason for the red color of the *P. alkekengi* fruit (Figure 1). Lutein contents were notably lower during the green ripening stage than during the other stages, showing a marked rise from this stage to the color conversion stage, hinting at a potential role in *P. alkekengi* fruit coloration. Significant discrepancies in β-carotene amounts were observed between the firm ripening and full ripening phases and between the green ripening and color conversion phases, showing the crucial function of β-carotene in *P. alkekengi* fruit pigmentation. During the S2–S3 stages, the lycopene contents in *P. alkekengi* fruits increased significantly and then experienced a slight decline in the S4 stage. This decrease could be attributed to the physiological and biochemical processes that lead to the synthesis of other compounds. Throughout the four stages of *P. alkekengi* fruit development, lycopene consistently maintained a higher content compared to lutein and β-carotene, suggesting that it played a substantial role in determining the fruit’s coloration and was a primary pigment responsible for its hue.

### 2.2. Transcriptome Analysis of P. pubescens and P. alkekengi Fruits at Different Developmental Stages

#### 2.2.1. Overview of Transcriptomic Data

Transcriptomic sequencings were conducted on 24 fruit samples collected from *P. pubescens* and *P. alkekengi* at four different phases of development (S1–S4) (Appendix A). A total of 70.53 Gb of clean data was produced for *P. pubescens*, resulting in an average of 5.83 Gb per sample. The average percentage of Q30 bases was 94.72%, and the GC contents ranged from 42% to 45%. The percentage of clean reads aligning to the reference genome varied from 65.19% to 72.75%.

A total of 69.9 Gb of clean data was obtained for *P. alkekengi* with an average of 5.82 Gb per sample. The average percentage of Q30 bases was 97.12%, and the range of GC contents varied between 42.26% and 43.26%. The Unigenes sequences from *P. alkekengi* were compared to gene sequences from several databases, including NCBI_nr, GO, KEGG, Pfam, Swiss-Prot, and Eggnog (Appendix A). As a result, a total of 59,393 genes were annotated in *P. alkekengi*.

Principal component analysis (PCA) was conducted to explore the intrinsic variations within the transcriptomes of *P. pubescens* and *P. alkekengi*. The PCA was anchored by the first (PC1) and second principal components (PC2), representing the principal and orthogonal directions of dataset variance, respectively. The scatter plot from PCA clearly separated the samples along the PC1 and PC2 axes, highlighting considerable variability among groups. The segregation and clustering patterns suggest significant genetic distinctions between the two species. The consistent distribution of biological replicates within clusters confirms the experiment’s reliability and the reproducibility of the results. The heatmaps illustrating Pearson correlation coefficients revealed values ranging from 0.7 to 1, with the majority surpassing 0.9, corroborating the robust clustering and high correlation among replicates. Notably, the correlation heatmap for both species demonstrates a strong consensus among biological replicates, validating the uniformity of sample categorization. The thorough examination of PCA and heatmap outcomes revealed that the RNA data exhibited satisfactory quality and was suitable for subsequent analyses (Figure 3).

#### 2.2.2. DEGs Identification

Comparative analysis disclosed a total of 7185 differentially DEGs in the adjacent developmental stages of *P. pubescens* fruits with 3503 genes downregulated and 3682 genes upregulated. The comparisons between YS2 and YS1, YS3 and YS2, and YS4 and YS3 resulted in 4175, 1601, and 1409 DEGs, respectively (Figure 4). Notably, the YS2 vs. YS1 comparison exhibited the maximum number of upregulated and downregulated DEGs, indicating a substantial change in gene expression at the YS2 stage.

A significant number of DEGs were observed in *P. alkekengi*, comprising 17,763 genes across the adjacent developmental stages. Of these, 8494 genes were upregulated, and 9269 genes were downregulated. The comparisons of RS2 vs. RS1, RS3 vs. RS2, and RS4 vs. RS3 revealed 5835, 7551, and 4377 DEGs, respectively (Figure 3). The comparison of RS3 and RS2 had the maximum number of DEGs, suggesting a significant change in gene expression during these stages.

In *P. pubescens*, 72 DEGs were identified as common to all four developmental stages through a Venn diagram analysis, while in *P. alkekengi*, there were 601 DEGs. These shared DEGs across various stages implied potential regulatory functions in the fruit’s color-changing process. Furthermore, heatmaps were generated for the top 100 genes in the three comparative groups of *P. pubescens* and *P. alkekengi* to improve the visualization of DEG distributions (Figure 5).

The GO enrichment analyses of the three comparison groups for *P. pubescens* revealed that the upregulated DEGs were predominantly linked to the cellular components of the nucleus (GO:0005634), comprising 12.64% of the total, and the chloroplast (GO:0009507), accounting for 11.61%. Significantly, the downregulated genes in all comparison groups were predominantly associated with the nucleus (GO:0005634). In the case of *P. alkekengi*, the nucleus (GO:0005634) was consistently identified as the most prominent cellular component across all three comparison groups. These findings suggested that nuclear function was conserved in gene regulation during fruit development.

KEGG pathway enrichment analyses were conducted to gain further insight into the metabolic pathways associated with these DEGs. In the comparison between groups YS2 and YS1, the pathways that were enriched in *P. pubescens* included biotin metabolism (map00780), carbon fixation in photosynthetic organisms (map00710), and carotenoid biosynthesis (map00906). In the comparison between groups YS3 and YS2, the pathways that were significantly enriched in *P. pubescens* were carotenoid biosynthesis (map00906), riboflavin metabolism (map00740), alpha-linolenic acid metabolism (map00592), and starch and sucrose metabolism (map00500). In the comparison between groups YS4 and YS3, the pathways enriched in *P. pubescens* included carotenoid biosynthesis (map00906), riboflavin metabolism (map00740), and starch and sucrose metabolism (map00500). A shift in focus to the comparison between groups RS2 and RS1 revealed that the pathways significantly enriched in *P. alkekengi* encompassed plant hormone signal transduction (map04075), biosynthesis of phenylpropanoids (map00940), flavonoid biosynthesis (map00941), the MAPK signaling pathway in plants (map04016), carotenoid biosynthesis (map00906), and anthocyanin biosynthesis (map00942). In the comparison between groups RS3 and RS2, the pathways enriched in *P. alkekengi* included plant hormone signal transduction (map04075), flavonoid biosynthesis (map00941), phenylpropanoids biosynthesis (map00940), and carotenoid biosynthesis (map00906). Lastly, the comparison between groups RS4 and RS3 revealed that the pathways significantly enriched in *P. alkekengi* were phenylpropanoids biosynthesis (map00940), plant hormone signal transduction (map04075), and flavonoid biosynthesis (map00941) (Figure 6).

Integrated analysis demonstrated that the carotenoid biosynthetic pathway (map00906) significantly influences fruit coloration in *P. pubescens*. In contrast, the carotenoid (map00906) and flavonoid biosynthetic pathways (map00941) significantly impacted fruit coloration in *P. alkekengi*. These findings indicate that these pathways are pivotal in orchestrating the complex molecular biological processes that affect fruit coloration.

#### 2.2.3. Uncovering the Core Genes in the Carotenoid Biosynthesis Pathway

Our study identified the carotenoid biosynthesis pathway (map00906) as the primary mechanism regulating fruit color development in *P. pubescens* and *P. alkekengi* using physiological pigment analysis and transcriptome sequencing. The phytoene synthase gene (PSY) was the first essential gene for the synthesis of phytoene to be isolated [22]. Utilizing FPKM values derived from transcriptome data and subsequent log10 transformation, heatmaps illustrating the carotenoid biosynthesis pathways in *P. pubescens* and *P. alkekengi* were generated (Figure 7). The expression levels of key genes in *P*. *pubescens* and *P*. *alkekengi* are shown in Table 7 and Table 8.

We identified two genes (pf01G007760 and pf05G128020) that regulated phytoene synthase in *P. pubescens* fruits. Our findings indicated that these genes exhibited elevated expression levels during the later stages of fruit development, particularly pf01G007760, demonstrating a notable surge in expression from S2 to S3. Subsequently, phytoene is subjected to catalysis by genes such as phytoene desaturase (PDS), ε-carotene isomerase (Z-ISO), ε-carotene desaturase (ZDS), and carotenoid isomerase (crtISO), resulting in the production of lycopene. Notably, the PDS, ZDS, and crtISO genes exhibited a general upward trend in expression. In the subsequent phase of lutein synthesis, the expression levels of lycopene ε-cyclase (LCYE), lycopene β-cyclase (LCYB), and β-carotene ε-hydroxylase (crtZ) genes all exhibited overall increases. In contrast, the transition from lycopene to β-carotene synthesis was accompanied by an increased expression of the LCYB gene from S1 to S4. Concurrently, the downstream gene CHYB decreased during stages S2 and S3, resulting in the accumulation of precursor substances.

Throughout the fruit developmental stages of *Physalis alkekengi*, the co-regulation of two genes (*TRINITY_DN17898_c1_g1* and *TRINITY_DN18612_c3_g2*) influenced phytoene synthase activity. Specifically, the gene *TRINITY_DN17898_c1_g1* demonstrated pronounced expression fluctuations, particularly when contrasting stages S3 and S2. Carotenoid isomerase, an enzyme pivotal to lycopene synthesis, exhibited significant upregulation as development progressed from stages S2 to S3. During the conversion of lycopene to δ-carotene, the gene responsible for lycopene ε-cyclase (*LCYE*) showed a declining expression pattern. This suggested a potential bottleneck in the pathway, leading to an accumulation of precursors. Concurrently, the pivotal genes for α-carotene synthesis exhibited an increasing expression trend with a marked upregulation at stages S2 and S3. In a parallel metabolic branch, the gene encoding β-carotene hydroxylase (*LCYB*), essential for β-carotene production, also displayed upregulation during this developmental transition.

#### 2.2.4. Key Transcription Factors (TFs) in the Orchestration of Fruit Maturation and Pigmentation

In this study, we identified a substantial number of TFs associated with developmental stage comparisons: 2826 in YS2 vs. YS1, 1087 in YS3 vs. YS2, and 903 in YS4 vs. YS3 for *P. pubescens*; and 867 in RS2 vs. RS1, 871 in RS3 vs. RS2, and 533 in RS4 vs. RS3 for *P. alkekengi*. The majority of these TFs were categorized into the bHLH, AP2/ERF, NAC, MYB, WRKY, FAR1, GRAS, and C2H2 families, which are known to play pivotal roles in plant development [23]. The expression levels of key transcription factors in *P. pubescens* and *P. alkekengi* are shown in Table 9 and Table 10.

Throughout the various fruit developmental stages of *P. pubescens*, a total of 10 TFs from families including AP2, bHLH, MYB-related, NAC, G2-like, HSF, WRKY, ERF, LBD, and MIKC-type MADS-domain were discovered to be implicated in the regulation of carotenoid biosynthesis. Among these were two key TFs, the MYB-related and bHLH TFs, which were found to regulate the synthesis of carotenoids critical for fruit development and coloration in *P. pubescens*. The MYB-related TF was observed to regulate four zeta-carotene isomerase genes (*pf01G023340*, *pf03G062680*, *pf06G156060*, and *pf09G225450*), exerting a positive regulatory effect on *pf01G023340* and adverse effects on the others. Conversely, the bHLH was identified as a regulator of the down-expression of the β-hydroxylase gene (pf07G199260), influencing carotenoid biosynthesis.

During the fruit developmental stages of *P. alkekengi*, six pivotal TFs (bHLH, HSF, WRKY, M-type MADS, AP2, and NAC) were identified that governed fruit coloration by modulating the expression of enzymes engaged in carotenoid synthesis. The bHLH regulated the genes encoding key enzymes such as beta-glucosidase (*TRINITY_DN17230_c1_g3*), capsanthin–capsorubin synthase (*TRINITY_DN1341_c0_g1* and *TRINITY_DN608_c0_g1*), and carotenoid epsilon-monooxygenase (*TRINITY_DN9566_c0_g1*). Notably, *TRINITY_DN1341_c0_g1* was upregulated, whereas the expression of the other genes was significantly downregulated. The HSF negatively regulated three beta-glucosidase genes, among which *TRINITY_DN17243_c4_g11* exhibited significant downregulation. The WRKY was implicated in the downregulation of another beta-glucosidase gene (*TRINITY_DN19440_c1_g1*). On the other hand, the M-type MADS upregulated the expression of two aldehyde dehydrogenase genes (*TRINITY_DN18484_c4_g3* and *TRINITY_DN19115_c4_g3*). The AP2 regulated the downregulation of *TRINITY_DN19671_c1_g1*, which inhibited the synthesis of beta-carotene isomerase. Furthermore, the NAC influenced the regulation of four capsanthin–capsorubin synthase genes with significant inhibitory effects on *TRINITY_DN18488_c1_g4* and a promoting effect on *TRINITY_DN5800_c0_g1*.

#### 2.2.5. Validation of Transcriptome Sequencing Data through qRT-PCR Analysis

In our study, six genes were randomly selected for qRT-PCR analysis to evaluate the expression levels of genes related to the color change in the fruit of *P. pubescens* and *P. alkekengi* at four different developmental stages. The results demonstrated that the expression levels of the genes detected by qRT-PCR were consistent with those from the sequencing, verifying the reliability of the transcriptome data (Figure 8).

## 3. Discussion

The coloration of plant fruits is primarily associated with three categories of pigments, including carotenoids, flavonoids (anthocyanins), and betalains. Rodrigo et al. have identified carotenoids as the primary pigments responsible for the yellow, orange, and red colors in mature citrus fruits [24]. Martel et al. suggested that lycopene and lutein contribute to the red and yellow coloring of tomato fruits [25]. A study conducted by Agnieszka et al. revealed that *Physalis peruviana* is abundant in carotenoids with α-carotene being the principal contributor [26]. Furthermore, a study by Wen et al. demonstrated that β-carotene primarily accounted for the yellow and orange hue in ripe yellow, orange, and red Physalis fruits [27]. This finding aligned with the results of our study, which identified β-carotene as the essential pigment responsible for the coloration of *P. pubescens* fruits and both β-carotene and lycopene as the primary pigments responsible for the coloration of *P. alkekengi* fruits. The coloration of plant fruits is jointly influenced by the types of pigments and their relative concentrations. In *Vaccinium corymbosum* cv. *Bluecrop*, the relative increase in delphinidin and malvidin is correlated with the darkening of the fruit peel color [28]. The accumulation of lycopene confers the red coloration to tomato fruit [29]. During the green ripening stage (S1), compared to *P. pubescens*, *P. alkekengi* exhibits a lower chlorophyll content in the fruit, yet its green color is more vivid. This could be attributed to the higher content of flavonoids and carotenoids in *P. alkekengi*, which enhances the fruit’s pigmentation effect due to the accumulation of these pigments.

### 3.1. Key Genes Driving Fruit Color Variation

Some studies suggest that the color differences between yellow and red fruits are due to the accumulation of carotenoids within the fruits [21]. In papaya fruit, carotenoids (lycopene, β-carotene, β-cryptoxanthin) accumulate extensively in the flesh; therefore, the flesh color changes from white to red [30]. In the current study, stages S3 and S4 were identified as pivotal periods for the variation in carotenoid content and differential fruit coloration in *P. pubescens* (with yellow fruits) and *P. alkekengi* (with orange–red fruits) (Figure 1, Table 3). Specifically, during the S3–S4 phase, *P. pubescens* primarily accumulates β-carotene, resulting in a yellow fruit appearance, whereas *P. alkekengi* accumulates both β-carotene and lycopene, leading to a red fruit coloration. The genes involved in carotenoid metabolism have been identified and investigated in several plants. It has been documented that a 2 bp insertion is present in the chromoplast-specific *LCYB*, resulting in a recessive loss-of-function mutation, so as to accumulate a large amount of lycopene which is responsible for the red flesh [31]. The silencing of the *AcBCH1* in kiwifruit results in decreased levels of zeaxanthin, lutein, and β-cryptoxanthin, while there is an increase in the content of β-carotene [32]. In the subsequent stages of fruit development in *P. pubescens*, the fruit displayed a gradual change in coloration. The key genes in the carotenoid biosynthesis pathway were *pf02G043370* (encoding β-carotene hydroxylase) and *pf06G178980* (lycopene β-cyclase), demonstrating a positive regulatory relationship with pigmentation. Specifically, *pf06G178980* encodes lycopene β-cyclase, which is an enzyme essential for cyclizing lycopene into β-carotene. Furthermore, it regulated the content of β-carotene by repressing the expression of the β-carotene hydroxylase gene.

Lycopene ε-cyclase (*LCYE*) is identified as the key determinant of differential carotenoid accumulation in the red flesh and yellow rind of papayas [30]. In orange melon fruits, higher expression levels of *LCYB* have been demonstrated to correlate with increased β-carotene content [33], which is consistent with the present study’s results. The essential genes involved in the development and coloration of *P. alkekengi* fruits were *TRINITY_DN20150_c1_g3* (lycopene β-cyclase), *TRINITY_DN10183_c0_g1* (carotenoid isomerase), and *TRINITY_DN23805_c0_g3* (lycopene ε-cyclase). As the fruit developed and ripened, *TRINITY_DN20150_c1_g3* (lycopene β-cyclase) was found to control the formation of β-carotene and was observed to be positively regulated. From the color transition stage (RS2) to the firm ripe stage (RS3), there was a notable increase in the expression of *TRINITY_DN10183_c0_g1* (carotenoid isomerase), which promoted the synthesis of lycopene. In the full ripening stage (RS4), the expressions of both *TRINITY_DN10183_c0_g1* (a carotenoid isomerase) and *TRINITY_DN23805_c0_g3* (lycopene ε-cyclase) were found to be downregulated with the latter displaying a notable decline in gene expression. At this juncture, the lycopene contents also exhibited a decline. Both genes demonstrated a positive correlation with lycopene contents; however, the downregulation of the latter gene contributed to the accumulation of its upstream metabolite, lycopene.

### 3.2. Regulation of Fruit Color by TFs

Recent advancements in the study of carotenoid synthetic metabolism have revealed that specific TFs play a crucial role in regulating the synthesis and metabolism of carotenoids. For example, the MYB family has been shown to regulate the synthesis of carotenoids in kiwifruit [34], and the BBX family has been shown to regulate carotenoid metabolism in tomatoes [35]. The NAC and MADS-box families have also been shown to control carotenoid production and metabolism [36,37]. In addition, the AP2/ERF family has been discovered to play a role in controlling carotenoid production and metabolism especially in response to external stress [38].

In *P. pubescens* fruit development, 10 TFs belonging to diverse families, such as AP2, bHLH, MYB-related, NAC, G2-like, HSF, WRKY, ERF, LBD, and MIKC-type MADS-domain, were identified as regulators of the carotenoid biosynthesis pathway. It is noteworthy that the gene *pf01G023340*, which was regulated by the MYB-related TF, exhibited a positive correlation with carotenoid contents. In contrast, genes *pf03G062680*, *pf06G156060*, and *pf09G225450*, which were also under the control of the MYB-related TF, and *pf07G199260*, which was controlled by bHLH, showed negative regulation. Furthermore, it was observed that the MYB-related TF exerted dual effects on zeta-carotene isomerase. Specifically, *pf01G023340* showed positive regulation, while *pf03G062680*, *pf06G156060*, and *pf09G225450* showed negative regulation. All of these genes encoded zeta-carotene isomerase and were modulated by the MYB-related TF.

Throughout the developmental stages of *P. alkekengi* fruits, six TFs were discovered to regulate the enzymes responsible for carotenoid synthesis. Among these TFs were bHLH, HSF, WRKY, M-type MADS, AP2, and NAC, each playing a specific role in controlling the synthesis of enzymes related to carotenoid production. For instance, bHLH, HSF, and WRKY were found to negatively regulate beta-glucosidase genes. M-type MADS controls aldehyde dehydrogenase. AP2 regulates beta-carotene isomerase, and bHLH notably influences carotenoid epsilon-monooxygenase. Additionally, NAC and bHLH were identified as regulators of capsanthin–capsorubin synthase. Notably, bHLH was found to regulate two distinct enzyme genes (*TRINITY_DN1341_c0_g1* and *TRINITY_DN9566_c0_g1*) with varying expression patterns within the carotenoid synthesis pathway of *P. alkekengi*. Collectively, these findings shed light on further exploration into carotenoid biosynthesis in both *P. pubescens* and *P. alkekengi*.

## 4. Materials and Methods

### 4.1. Plant Materials

The botanical specimens employed in the investigation comprised fruits derived from two discrete species within the Physalis genus. *P. pubescens*, which exhibited yellow fruits, and *P. alkekengi*, which displayed orange–red fruits. All plants were cultivated from March to August 2021 at the Vegetable Research and Teaching Base of Jilin Agricultural University, Jilin Province, China (latitude: 43.817, longitude: 125.406; elevation: 217 m). Each research field was populated with 20 plants, arranged with a row spacing of 150 cm and a plant spacing of 45 cm, based on a randomized block design. Samples of fruit from *P. pubescens* and *P. alkekengi* were collected at four distinct stages of fruit development, each following the same number of days post-anthesis (DPA): the S1 green ripening phase (15 DPA), the S2 color conversion phase (25 DPA), the S3 firm ripening phase (35 DPA), and the S4 full ripening phase (45 DPA). The entire fruit samples comprising the skin and flesh were rapidly frozen in liquid nitrogen upon collection and subsequently preserved at −80 °C for constructing transcriptome libraries and validation through quantitative reverse transcription polymerase chain reaction (qRT-PCR). The experiment was conducted with three biological replicates per stage, with each replicate comprising 3 fruits, resulting in a total of 24 samples.

### 4.2. Determination of Pigment Contents of Fruits

#### 4.2.1. Quantification of Total Flavonoid Contents

The fresh fruit sample was crushed and placed in a container. A 40 mL 70% methanol solution was added, which was followed by ultrasonic treatment (power 120 W, frequency 40 kHz) for 30 min. The mixture was then filtered, evaporated, and dissolved in 70% (*v*/*v*) ethanol, and the final volume was adjusted to 10 mL. A 2 mL aliquot was combined with 6 mL of water and 1 mL of a 5% (*w*/*v*) sodium nitrite solution, and the resulting mixture was allowed to stand for 6 min. Subsequently, 1 mL of a 10% (*w*/*v*) aluminum nitrate solution was added to the mixture, thoroughly mixed, and allowed to stand for another 6 min. Finally, 10 mL of sodium hydroxide solution was added. The mixture was diluted to the mark with water, thoroughly mixed, and allowed to stand for 15 min.

A spectrophotometer (T6 New Century, Beijing Puxi General Instrument Co., Ltd., Beijing, China) was used to quantify total flavonoids following the operating protocols outlined in the instructional guide. The absorbance of the extracted solution to be quantified was measured at a wavelength of 510 nm, which is a commonly utilized wavelength for flavonoid quantification assays [39,40]. The total flavonoid content was determined by employing the following formula:$$\mathrm{Total}\;\mathrm{flavonoid}\;\mathrm{content}\;(\mathrm{mg}/g)=\frac{C\times V}{G}$$
where C is the concentration of flavonoids in the fresh fruit sample, which is determined from the standard curve. V is the volume of the extracted solution, which is 2 mL, and G is the weight of the fresh fruit sample in grams.

#### 4.2.2. Extraction and Determination of Total Anthocyanin Contents

A quantity of 0.10 g of fresh fruit was extracted using 10 mL of 95% ethanol containing 0.1 M HCl. The extraction process was conducted at 60 °C, with the procedure being repeated twice, which was followed by an incubation period of 1 h. Subsequently, the resulting extracts were combined to achieve a final volume of 25 mL, comprising an ethanolic solution with 0.1 M HCl.

The total anthocyanin contents of the fruit samples were quantified following the methodology delineated in the operating instructions. The absorbance readings of the samples were obtained at wavelengths of 520, 620, and 650 nm using a spectrophotometer (T6 New Century, Beijing Puxi General Instrument Co., Ltd., Beijing, China) [19]. The calculation of total anthocyanins was performed using the following formula:$$\mathrm{Total}\;\mathrm{anthocyanin}\;\mathrm{content}\;Q\;(\mathrm{mmol}/g\;\mathrm{FW})=\frac{A_{\lambda }\times V\times 1000}{489.72M}$$

The formula A_λ_ = (A_530_ − A_620_) − 0.1 (A_650_ − A_620_) is used to calculate absorbance values in spectrophotometry. Here, the A_530_, A_620_, and A_650_ represent the absorbance readings at 530 nm, 620 nm, and 650 nm, respectively. V is the volume of the extraction solution in milliliters, while M indicates the weight of the fresh sample in grams. FW stands for fresh weight.

#### 4.2.3. Measurement of Chlorophyll Contents

The extraction of chlorophyll from the fruit samples was performed as previously described [41]. Weighed out at 0.30 g, the freshly chopped fruit was placed into a 15 mL centrifuge tube which had been wrapped with aluminum foil. A 1:1 volume-to-volume mixture of 95% ethanol and 80% acetone was added to it to reach a final volume of 10 mL. The mixture was then incubated in a dark environment at 30 °C with intermittent shaking until the green color of the sample had completely disappeared.

Following the operating procedures outlined in the instruction manual for determining chlorophyll, absorbance readings at 665 and 649 nm were measured for the extracted solution using a spectrophotometer (T6 New Century, Beijing Puxi General Instrument Co., Ltd., Beijing, China) [41]. The calculation formula for calculating the concentrations of chlorophyll a (C_a_) and chlorophyll b (C_b_) and the total chlorophyll content (C) was as follows:
$$\begin{array}{c} C_{a}=13.95\times A_{665}-6.88\times A_{649}\\ C_{b}=24.96\times A_{649}-7.32\times A_{665}\\ \mathrm{Total}\;\mathrm{Chlorophyll}\;(\mathrm{mg}/L)\;=C_{a}+C_{b}\\ \mathrm{Total}\;\mathrm{Chlorophyll}\;\mathrm{Content}\;C\;(\mathrm{mg}/g\;\mathrm{FW})=\frac{Total\;Chlorophyll\;(mg/L)\times V\times D}{g} \end{array}$$

A_665_, A_649_, and A_665_ represent the absorbance readings at 665 nm, 649 nm, and 665 nm, respectively. Ca is the concentration of chlorophyll a in mg/L; Cb is the concentration of chlorophyll b in mg/L; V represents the volume of the extracted solution measured in milliliters (mL); D is used to indicate the dilution factor; and g signifies the fresh weight of the sample in grams (g). FW stands for fresh weight.

#### 4.2.4. Carotenoid Quantification

A 0.1 g sample of freshly chopped fruit was placed in a mortar, 1 mL of distilled water and a small amount of Reagent One were added, and the mixture was thoroughly crushed under dim light. The mortar was then rinsed with 80% acetone, and the solution obtained was transferred to a 10 mL centrifuge tube and adjusted to a final volume of 10 mL. The tube was wrapped in aluminum foil to prevent light exposure and extracted for 3 h with periodic inversions to mix (three total inversions during extraction) until the remaining tissue at the bottom was nearly white.

Following the instructions detailed in the user manual of the carotenoid assay kit (product number; Boxbio, Beijing, China), a spectrophotometer (T6 New Century, Beijing Puxi General Instrument Co., Ltd., Beijing, China) was employed for the quantification of absorbance readings at 440 nm as mentioned in a previous report [42]. The calculation of carotenoid content was based on the formula provided below:$$\mathrm{Carotenoid}\;\mathrm{Content}\;(\mathrm{mg}/g)=\frac{A_{440}\times V_{ext}\times 1000\times D}{\varepsilon \times d\times W}=\frac{0.04\times A_{440}\times D}{W}$$

A_440_ represents the absorbance reading at 440 nm. The term V_ext_ denotes the extracted sample’s total volume, which is equivalent to 10 mL. The symbol ε represents the specific empirical extinction coefficient for carotenoids, which was set at 250 L/g/cm. This coefficient is utilized to establish a correlation between the sample’s absorbance and the concentration of carotenoids. The variable d denotes the optical path length of the cuvette, which has been standardized at 1 cm. Variable D represents the dilution factor, compensating for any dilution during sample preparation. The symbol W represents the mass of the sample in grams (g), which is used to measure the carotenoid content per gram of the sample. To convert the concentration from milligrams per liter to milligrams per gram when calculating based on the sample’s weight, the factor 1000 is utilized.

#### 4.2.5. HPLC-Based Quantification of Lycopene, β-Carotene and Lutein in Fruit Samples

A 0.5 g sample of homogenized dry fruit powder was precisely weighed and transferred to a 15 mL brown centrifuge tube. To this was added 5 mL of a 0.1% BHT-ethanol solution, which was followed by thorough mixing and shaking to ensure homogeneity. The tube was then placed on a 25 °C constant temperature shaker and oscillated in the dark at room temperature for 4 h to complete the extraction. The supernatant was then transferred to another brown centrifuge tube, and the extraction process was repeated. The supernatants from both extractions were combined, shaken to ensure thorough mixing, and filtered through a 0.22 µm filter membrane into a 2 mL brown injection vial for the determination of lycopene, β-carotene, and lutein content.

Following the extraction protocols for lycopene, β-carotene and lutein [43,44,45], we utilized the Rigol L3000 HPLC system (RIGOL Technologies, Inc., Beijing, China), equipped with a VWD C18 column (250 mm × 4.6 mm, 5 μm), to determine their respective contents.

### 4.3. Transcriptome Sequencing

Total RNA samples were obtained from the whole fruits of *P. pubescens* (Y) and *P. alkekengi* (R) at various stages of ripening (S1–S4) using the Trizol reagent kit (Invitrogen, Carlsbad, CA, USA). A preliminary assessment of RNA integrity and possible contamination was performed using agarose gel electrophoresis. The RNA quality was assessed by measuring the A260/A280 ratio using a NanoDrop spectrophotometer ND-1000 (Manufacturer information), while the concentration was calculated using a Qubit fluorometer (Manufacturer information). In addition, the Bioanalyzer 2100 (Agilent, Santa Clara, CA, USA) was used to verify the integrity of the RNA [46,47,48]. Following the manufacturer’s instructions, cDNA libraries for *P. pubescens* were prepared using a general method, as a reference genome was available. Conversely, cDNA libraries for *P. alkekengi* were de novo constructed due to the absence of a reference genome. Upon completion of library quality validation, the Illumina NovaSeq™ 6000 platform (LC Bio Technology Co., Ltd. Hangzhou, China) was utilized for conducting paired-end sequencing with the read length set at 150 bp, following the established standard operating procedures [49]. Subsequently, the software Cutadapt [50] was utilized to remove adapter sequences and low-quality reads to obtain high-quality data.

For *P. pubescens*, the clean reads were aligned to its reference genome, which was obtained from the National Genomics Data Center (NGDC) at https://ngdc.cncb.ac.cn/gwh/Assembly/9765/show (accessed on 15 October 2022). Hisat2 [51] was employed for sequence alignment, and then Stringtie [52,53] was utilized to reconstruct transcripts and estimate the expression levels of all genes in each sample. For *P. alkekengi*, the Trinity software [54] was used to perform de novo transcriptome assemblies and generate unigenes. Subsequently, the unigenes were aligned to six databases, namely NCBI_NR (https://www.ncbi.nlm.nih.gov/protein, accessed on 8 October 2022), Gene Ontology (GO) (https://www.geneontology.org/, accessed on 8 October 2022), Kyoto Encyclopedia of Genes and Genomes (KEGG) (https://www.kegg.jp/, accessed on 8 October 2022), Pfam (https://www.ebi.ac.uk/interpro/, accessed on 8 October 2022), Swiss-Prot (https://www.uniprot.org/, accessed on 9 October 2022), and Eggnog (http://eggnog5.embl.de/, accessed on 9 October 2022), using the DIAMOND software (https://www.ebi.ac.uk/Tools/search/diamond/, accessed on 9 October 2022). The expression patterns of the genes in all samples were quantified collectively and subjected to statistical analysis. Following normalization, gene expression levels were evaluated using the fragments per kilobase of transcript per million mapped reads (FPKM) approach. Differentially expressed genes (DEGs) were identified based on fold change and statistical significance. DEGs were identified using a fold change cutoff of greater than 2 (log2FC ≥ 1) and a *q*-value cutoff of less than 0.05 (representing the corrected *p*-value). The DESeq2 R package (3.6.2.) [55] was employed to analyze differential gene expression. The expression patterns of the genes that exhibited differential expression were displayed using heatmaps generated through hierarchical clustering. Moreover, GO and KEGG pathway enrichment analyses were conducted on DEGs using GOseq 1.34.1 and KOBAS 2.0 software (https://www.bioconductor.org/packages/release/bioc/html/goseq.html, accessed on 20 October 2022 and http://bioinfo.org/kobas, accessed on 23 October 2022). The LC Bio Cloud Platform (https://www.omicstudio.cn./home, accessed on 29 October 2022) was employed to generate Venn diagrams, volcano plots, and heatmaps.

### 4.4. Transcriptome Sequencing Validation through qRT-PCR

Six genes were selected randomly for validation via qRT-PCR to assess their expression patterns in the fruits of *P. pubescens* and *P. alkekengi* across four distinct developmental stages. The PCR primers were developed utilizing both the NCBI database (https://www.ncbi.nlm.nih.gov/tools/primer-blast/, accessed on 2 December 2022) and Primer 5.0 software (Premier Biosoft International, San Francisco, and California) and subsequently synthesized by Jilin Kume Biotech Co., Ltd. (Changchun, China). The specific primer sequences can be found in Table 11 and Table 12 with *PpGAPDH* serving as the internal control gene [56,57]. The average Ct value was standardized in relation to the internal control gene, and the 2^−∆∆Ct^ technique was employed to investigate the relative levels of gene expression.

### 4.5. Data Analysis

All samples were analyzed in triplicate unless stated otherwise. Data were reported as the mean ± standard deviation (SD). The processing of phenotypic data and graphing of fruit pigments were conducted using Microsoft Excel 2010 (Microsoft Corporation, Redmond, and Wash) and Microsoft Word 2010 (Microsoft Corporation, Redmond, and Wash). The normality and homogeneity of variances were examined by Shapiro–Wilk’s and Levene’s test (*p* < 0.05), respectively. The ANOVA statistical analysis was performed using DPS7.5 (Hangzhou Ruifeng Information Technology Co., Ltd., Hangzhou, China) at a significance level of *p* < 0.05 unless stated otherwise. Duncan’s multiple range tests were used for the comparison of multiple samples, and an independent-samples t-test was conducted for comparing outcomes between *P. pubescens* and *P. alkekengi* samples using online cloud platform tools (https://cloud.tencent.com/product/bi, accessed on 10 March 2022).

## 5. Conclusions

β-carotene was the primary pigment responsible for the mature hue of *P. pubescens*, while the fruit coloration of *P. alkekengi* was influenced by both lycopene and β-carotene. The primary metabolic pathway responsible for fruit pigmentation in *P. pubescens* and *P. alkekengi* was identified as carotenoid biosynthesis (map00906). During the stages of fruit growth and coloration in *P. pubescens*, the pivotal genes responsible for carotenoid production were pf02G043370 and pf06G178980. The production of zeta-carotene isomerase and β-hydroxylase was found to be controlled by two critical TFs, the MYB-related and bHLH TFs, respectively. The MYB-related TF has been demonstrated to exert positive and negative regulatory effects on zeta-carotene isomerase activity. The core genes involved in the production of carotenoids in *P. alkekengi* fruit growth and coloration were identified as TRINITY_DN20150_c1_g3, TRINITY_DN10183_c0_g1, and TRINITY_DN23805_c0_g3. The production of enzymes linked to carotenoid synthesis in *P. alkekengi* fruits was discovered to be under the regulation of six TFs, encompassing bHLH, HSF, WRKY, M-type MADS, AP2, and NAC. These findings underscored the complex regulatory network that governs pigmentation in these two Physalis species.

## Figures and Tables

**Figure 1 ijms-25-09852-f001:**
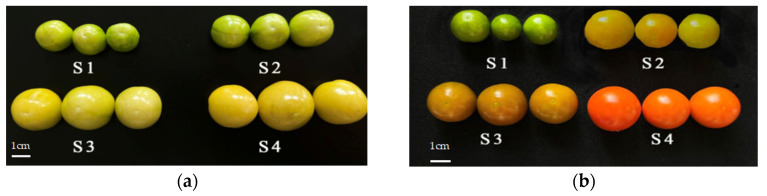
The coloration changes of (**a**) *P. pubescens* and (**b**) *P. alkekengi* fruits were observed at four specific stages of fruit development, which were each characterized by a specific number of days post-anthesis (DPA): the S1 green ripening phase (15 DPA), the S2 color conversion phase (25 DPA), the S3 firm ripening phase (35 DPA), and the S4 full ripening phase (45 DPA). The scale bar = 1 cm.

**Figure 2 ijms-25-09852-f002:**
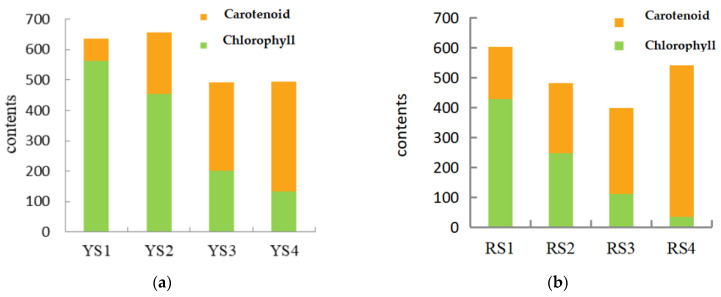
Chlorophyll-to-carotenoid content ratios across the four fruit developmental stages in (**a**) *P. pubescens* and (**b**) *P. alkekengi*.

**Figure 3 ijms-25-09852-f003:**
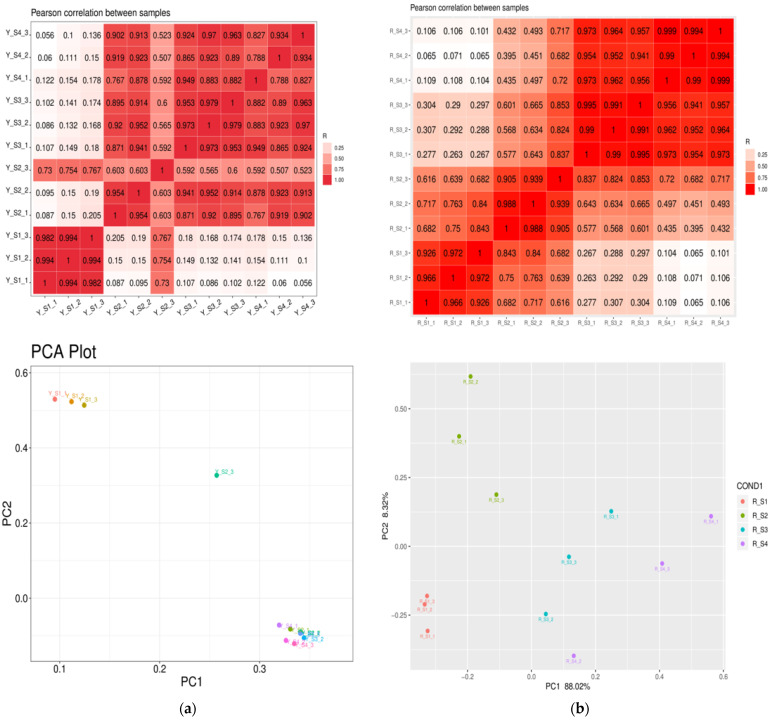
Pearson correlation heatmaps and PCA plots for (**a**) *P. pubescens* and (**b**) *P. alkekengi*.

**Figure 4 ijms-25-09852-f004:**
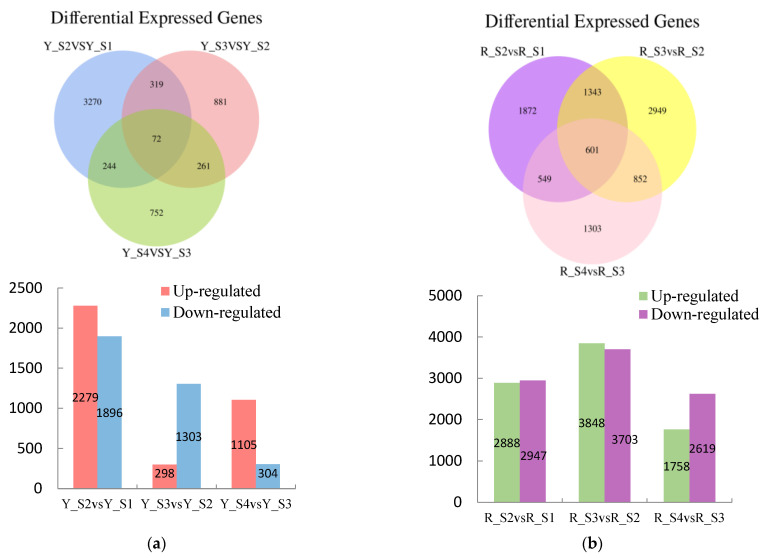
Venn diagrams of DEGs and the numbers of upregulated and downregulated DEGs in (**a**) *P. pubescens* and (**b**) *P. alkekengi*.

**Figure 5 ijms-25-09852-f005:**
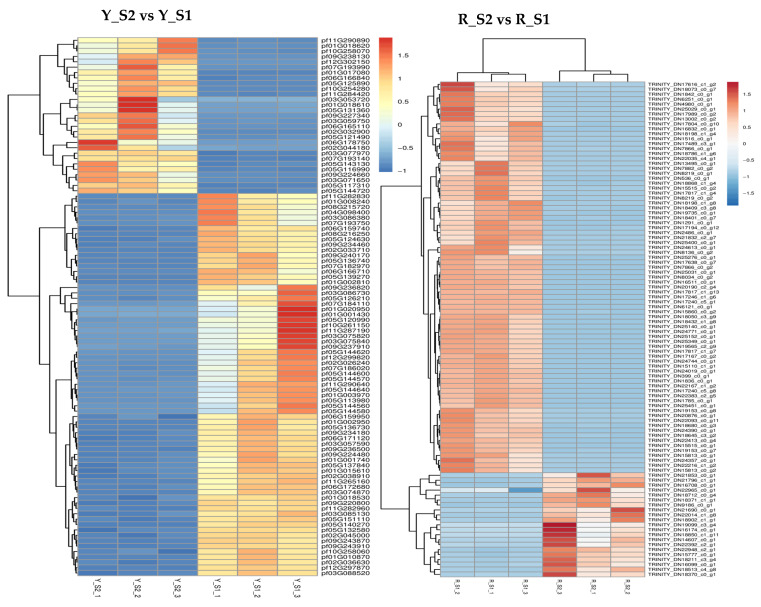
Heatmaps of hierarchical clustering of the top 100 DEGs in different comparison groups of (**a**) *P. pubescens* and (**b**) *P. alkekengi*.

**Figure 6 ijms-25-09852-f006:**
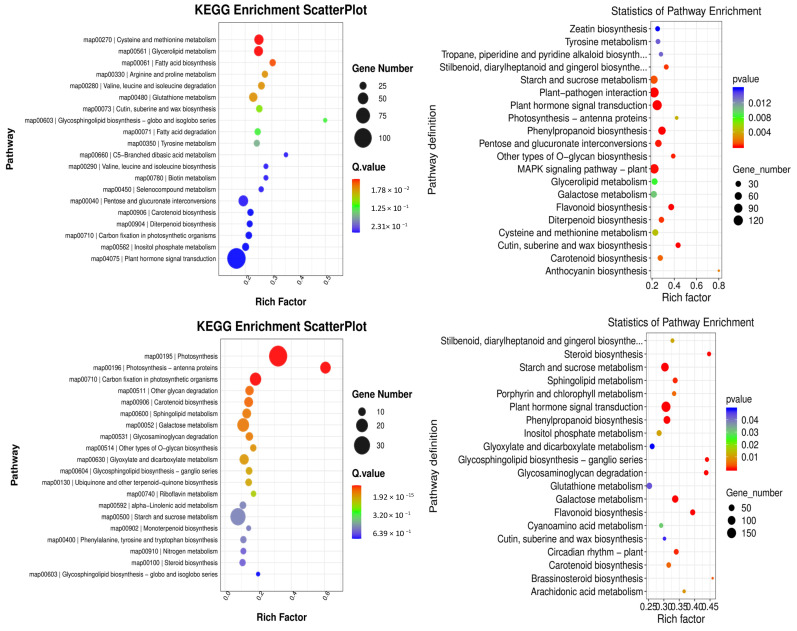
Scatter plots of KEGG enrichment analysis in different comparison groups of (**a**) *P. pubescens* and (**b**) *P. alkekengi*.

**Figure 7 ijms-25-09852-f007:**
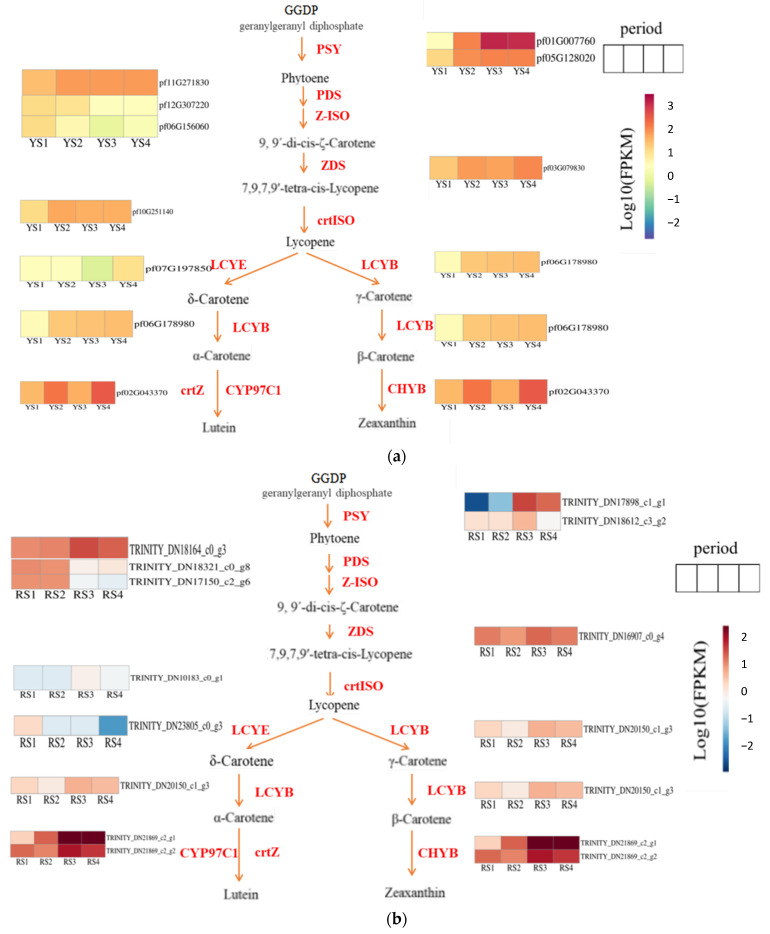
Expression heatmaps of essential metabolic genes in the carotenoid synthesis pathway for (**a**) *P. pubescens* and (**b**) *P. alkekengi* fruits.

**Figure 8 ijms-25-09852-f008:**
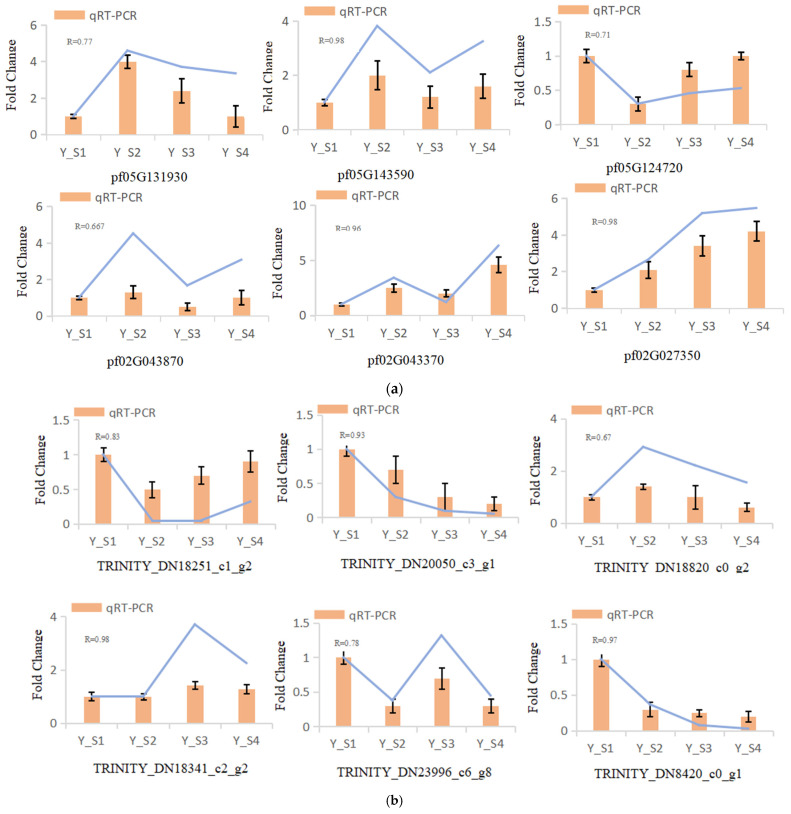
qRT-PCR validation of DEGs across four developmental stages (S1–S4) in (**a**) *P. pubescens* and (**b**) *P. alkekengi* fruits.

**Table 1 ijms-25-09852-t001:** Total flavonoid contents across the four fruit development stages in *P. pubescens* and *P. alkekengi*.

Material	Total Flavone Content (μg/g FW)			
	Green Ripening Stage (S1)	Color Conversion Stage (S2)	Firm Ripening Stage (S3)	Full Ripening Stage (S4)
*P. Pubescens* (Y)	112.64 ^c^ ± 0.02	247.42 ^b^ ± 0.15	333.58 ^a^ ± 0.01	341.65 ^a^ ± 0.11
*P. alkekengi* (R)	194.70 ^c^ ± 0.03	232.53 ^b^ ± 0.21	279.83 ^a^ ± 0.23	339.36 ^a^ ± 0.02

Note: Different letters (^a^, ^b^, and ^c^) denote significant differences at *p* < 0.05. FW stands for fresh weight.

**Table 2 ijms-25-09852-t002:** Anthocyanin contents across the four fruit development stages in *P. pubescens* and *P. alkekengi*.

Material	Anthocyanin Content (nmol/g FW)			
	Green Ripening Stage (S1)	Color Conversion Stage(S2)	Firm Ripening Stage (S3)	Full Ripening Stage (S4)
*P. pubescens* (Y)	11.03 ^c^ ± 0.12	21.62 ^b^ ± 0.03	30.83 ^a^ ± 0.06	32.75 ^a^ ± 0.18
*P. alkekengi* (R)	25.98 ^b^ ± 0.09	33.02 ^b^ ± 0.14	41.38 ^ab^ ± 0.21	58.73 ^a^ ± 0.04

Note: Different letters denote the significance levels of mean differences among groups at *p* < 0.05. FW stands for fresh weight.

**Table 3 ijms-25-09852-t003:** Carotenoid contents across the four fruit development stages in *P. pubescens* and *P. alkekengi*.

Material	Carotenoid Content (μg/g FW)			
	Green Ripening Stage (S1)	Color Conversion Stage (S2)	Firm Ripening Stage (S3)	Full Ripening Stage (S4)
*P.pubescens* (Y)	72.73 ^c^ ± 0.04	202.28 ^bc^ ± 0.02	289.25 ^b^ ± 0.23	360.07 ^a^ ± 0.15
*P. alkekengi* (R)	175.07 ^c^ ± 0.01	234.04 ^b^ ± 0.01	286.16 ^ab^ ± 0.03	506.81 ^a^ ± 0.11

Note: Different letters denote the significance levels of mean differences among groups at *p* < 0.05. FW stands for fresh weight.

**Table 4 ijms-25-09852-t004:** Chlorophyll contents across the four fruit development stages in *P. pubescens* and *P. alkekengi*.

Material	Chlorophyll Content (μg/g FW)			
	Green Ripening Stage (S1)	Color Conversion Stage (S2)	Firm Ripening Stage (S3)	Full Ripening Stage (S4)
*P.pubescens* (Y)	561.89 ^a^ ± 0.12	453.81 ^ab^ ± 0.01	202.36 ^bc^ ± 0.21	134.33 ^c^ ± 0.15
*P. alkekengi* (R)	427.87 ^a^ ± 0.02	246.66 ^ab^ ± 0.03	111.23 ^ab^ ± 0.13	34.96 ^bc^ ± 0.04

Note: Different letters denote the significance levels of mean differences among groups at *p* < 0.05. FW stands for fresh weight.

**Table 5 ijms-25-09852-t005:** Lutein, β-carotene, and lycopene contents across the four fruit development stages of *P. pubescens*.

Group	Lutein Content (μg/g FW)	β-Carotenoid Content (μg/g FW)	Lycopene Content (μg/g FW)
green ripening stage (YS1)	3.2614 ^a^ ± 0.01	0.4418 ^b^ ± 0.11	21.8701 ^a^ ± 0.13
color conversion stage (YS2)	0.7463 ^b^ ± 0.04	0.0320 ^b^ ± 0.12	6.1324 ^b^ ± 0.02
firm ripening stage (YS3)	0.0172 ^c^ ± 0.21	9.8345 ^a^ ± 0.21	4.6272 ^b^ ± 0.04
full ripening stage (YS4)	0.1598 ^c^ ± 0.15	10.9035 ^a^ ± 0.06	2.9403 ^b^ ± 0.11

Note: Different letters denote the significance levels of mean differences among groups at *p* < 0.05. FW stands for fresh weight.

**Table 6 ijms-25-09852-t006:** Lutein, β-carotene, and lycopene contents across the four fruit development stages of *P. alkekengi*.

Group	Lutein Content (μg/g FW)	β-Carotenoid Content (μg/g FW)	Lycopene Content (μg/g FW)
green ripening stag (RS1)	0.47 ^b^ ± 0.15	0.12 ^c^ ± 0.04	5.67 ^c^ ± 0.10
color conversion stage (RS2)	2.76 ^a^ ± 0.01	1.79 ^bc^ ± 0.13	55.67 ^ab^ ± 0.09
firm ripening stage (RS3)	2.43 ^a^ ± 0.11	10.42 ^b^ ± 0.22	63.45 ^a^ ± 0.04
full ripening stage (RS4)	2.31 ^a^ ± 0.03	37.00 ^a^ ± 0.11	42.85 ^b^ ± 0.02

Note: Different letters denote the significance levels of mean differences among groups at *p* < 0.05. FW stands for fresh weight.

**Table 7 ijms-25-09852-t007:** Expression levels of key genes in *P. pubescens* fruits.

Gene ID	Description	Y_S1 FPKM	Y_S2 FPKM	Y_S3 FPKM	Y_S4 FPKM
pf01G007760	Phytoene synthase	3.47	69.59	520.66	487.31
pf05G128020	Phytoene synthase 2	12.43	59.84	71.29	71.24
pf11G271830	Phytoene desaturase	22.15	48.57	44.07	43.95
pf12G307220	15-cis-zeta-carotene isomerase	11.78	8.70	2.99	3.69
pf03G079830	Zeta-carotene desaturase	18.47	47.10	41.42	63.49
pf10G251140	Prolycopene isomerase	11.08	34.50	32.84	34.04
pf06G178980	Lycopene beta cyclase	3.88	17.64	21.46	22.74
pf02G043370	Beta-carotene hydroxylase 1	26.45	90.02	31.68	166.98

**Table 8 ijms-25-09852-t008:** Expression levels of key genes in *P. alkekengi* fruits.

Gene ID	Description	Y_S1 FPKM	Y_S2 FPKM	Y_S3 FPKM	Y_S4 FPKM
TRINITY_DN17898_c1_g1	Phytoene synthase	0.07	0.58	42.23	26.61
TRINITY_DN18612_c3_g2	Bifunctional 15-cis-phytoene synthase	3.24	3.22	7.93	1.61
TRINITY_DN18164_c0_g3	Phytoene desaturase	16.02	17.54	37.98	27.25
TRINITY_DN16907_c0_g4	Zeta-carotene desaturase	19.76	13.14	25.47	18.46
TRINITY_DN10183_c0_g1	Prolycopene isomerase 1	0.77	0.70	1.77	1.49
TRINITY_DN23805_c0_g3	Lycopene epsilon cyclase	3.97	0.74	0.78	0.09
TRINITY_DN20150_c1_g3	Lycopene beta cyclase	4.13	2.28	8.66	6.84
TRINITY_DN21869_c2_g2	Beta-carotene hydroxylase 1	25.92	16.63	84.30	49.81

**Table 9 ijms-25-09852-t009:** Expression levels of key transcription factors in *P. pubescens* fruits.

Gene ID	Description	Y_S1 FPKM	Y_S2 FPKM	Y_S3 FPKM	Y_S4 FPKM
pf03G061540	Transcription factor MYB57	15.55	8.29	3.98	5.77
pf07G183140	Transcription factor MYB25	6.67	2.99	1.67	1.19
pf08G203960	Transcription factor MYB48	12.86	24.11	6.67	4.47
pf11G282100	Transcription factor MYB32	19.29	4.39	1.75	1.47
pf11G292660	Transcription factor bHLH66	7.14	6.28	1.81	0.63
pf01G004050	Transcription factor bHLH62	19.57	13.54	6.69	6.03
pf03G069270	Transcription factor bHLH49	3.70	2.17	8.42	25.28
pf03G081450	Transcription factor bHLH13	8.17	10.36	7.87	9.29
pf05G146670	Transcription factor bHLH93	18.65	13.59	4.80	4.32

**Table 10 ijms-25-09852-t010:** Expression levels of key transcription factors in *P. alkekengi* fruits.

Gene ID	Description	Y_S1 FPKM	Y_S2 FPKM	Y_S3 FPKM	Y_S4 FPKM
TRINITY_DN18784_c1_g5	Transcription factor bHLH79	16.20	13.33	5.51	3.00
TRINITY_DN11161_c0_g1	Transcription factor bHLH106	0.74	1.78	0.28	0.25
TRINITY_DN19371_c1_g6	Heat shock factor protein HSF8	15.34	16.33	19.02	26.89
TRINITY_DN17987_c4_g1	WRKY transcription factor 2	37.68	17.18	14.84	13.42
TRINITY_DN18956_c1_g8	WRKY transcription factor 6	2.42	5.26	5.30	4.09
TRINITY_DN21261_c0_g2	MADS-box transcription factor 23 isoform X1	40.57	35.88	11.06	12.78
TRINITY_DN21284_c1_g11	Ethylene-responsive transcription factor RAP2-13	3.33	9.01	3.63	0.49
TRINITY_DN19991_c0_g6	ethylene-responsive transcription factor RAP2-12	91.16	48.24	48.66	41.65
TRINITY_DN22588_c3_g4	NAC1 transcription factor	1.01	0.57	2.26	0.91

**Table 11 ijms-25-09852-t011:** The PCR primer sequences of *P. pubescens*.

Gene ID	Forward Primer (5′-3′)	Reverse Primer (5′-3′)
pf05G131930	GCTTGACTGGGCTGACCTCTTT	CTCTGGTTGTGGACATGGTGGA
pf05G143590	TCGCCTAACGAGAGGTGCTTCA	GATCAATGGCACCACCGCTGTT
pf05G124720	TGTGGCGTCACCCAGAGAATGT	CAGCAGGAGTCACGTTGGCAAT
pf02G043870	TGGCTTGGCTCGAACTCCTCAA	GCATCACGCATGGGTGGGTATC
pf02G043370	TTCCGCCACAGTCCGTTTCTTG	CTACAGCAGCACCAACCGAGAG
pf02G027350	GAGGCTTCTCGCTGAGGTCTCA	GGCTGAGGAGGCGTGACATCTA
Pp.GAPDH	TGTGGGTGTCAACGAGAAGGAATAC	ATAAGACCCTCCACAATGCCAAACC

**Table 12 ijms-25-09852-t012:** The PCR primer sequences of *P. alkekengi*.

Gene ID	Forward Primer(5′-3′)	Reverse Primer(5′-3′)
TRINITY_DN18251_c1_g2	TGGTTTGCCCATTAGGCTGT	TGATGGCTGCCTTGAGATCC
TRINITY_DN20050_C3_g1	GTCCACTCACTGCACCTTCA	CGTGAATCTCCACCGTCGAT
TRINITY_DN18820_c0_g2	TCATGAGGGTGCAATGCT	GTTATTGCTCCAGCCAACGC
TRINITY_DN18341_C2_g2	ACTCTGCTTGTGTGGTTGGT	TCCCTCCTCGTCATCCTGTT
TRINITY_DN23996_c6_g8	TCAGCCCATGGAGGAAATCG	CAGGACCAACACCAGAGCTT
TRINITY_DN8420_c0_g1	GCCCTCTCTGGAGCACATAC	GGGCACCTCTCTTGGAGTTC

## Data Availability

All RNA-Seq raw sequence data files are available in the NCBI SRA Database (https://dataview.ncbi.nlm.nih.gov) (accessed on 23 July 2024) under accession numbers SRR29923359, SRR29923358, SRR29923360, SRR29923361, SRR29923365, SRR29923362, SRR29923363, SRR29923364, SRR29923366, SRR29923367, SRR29923368, SRR29923369, SRR29923349, SRR29923356, SRR29923355, SRR29923354, SRR29923357, SRR29923353, SRR29923352, SRR29923351, SRR29923350, SRR29923348, SRR29923347, and SRR29923346.

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
