# Peer review of "Unraveling the Genetic Control of Pigment Accumulation in Physalis Fruits"

_ijms, 2024, doi:10.3390/ijms25189852_

Round 1

Reviewer 1 Report

Comments and Suggestions for Authors

The manuscript presents a clear research approach and employs mature analytical methods. However, the article lacks comparative analysis of the content of substances and gene expression levels between the two species during a single developmental stage. Additionally, the discussion section is somewhat lacking.

Major Comments:

1. From the phenotypic images, the S1 stage fruit of P. alkekengi appears greener than that of P. pubescens, yet the chlorophyll levels show an opposite characteristic. The discussion should include corresponding content to address this discrepancy.

2. The analyses in Result 2.1 only explore the comparison between different stages within each species, lacking comparison between different species during the same developmental stage. It is recommended to juxtapose morphological photographs, correlating the differences in key substances with the corresponding stages, and add comparative analysis between the two species.

3. For the transcriptome data analysis, please explain why the same reference genome set is not used? Utilizing the same reference genome facilitates comparison between the two species.

4. In Result 2.2.4, it is suggested to include a network interaction Figure of genes and transcription factors to visualize the results.

5. How were the two/three key genes identified in the two species? What are the differences in the expression levels of phytoene synthase between the two species?

6. The results mention the identified key genes and transcription factors, but the description of their expression levels is not intuitive enough. It is recommended to add corresponding figures or tables (supplementary figures or tables).

7. The discussion section needs further enhancement. The discussion lacks an integrated discussion of the changes in substance content and key gene expression levels within the fruit. The discussion should appropriately reference figures and tables from the results, combining research progress with the findings.

Minor Comments:

1. It is recommended to place Figure 8 in the Results section for easier comparison of phenotype with specific data.

2. The Latin names of the species in the header of Table 1 should be in italics.

3. There is Chinese text in Figure 1.

4. Figure 2 is distorted due to compression.

5. The font in Figure 4 is too small to be legible.

Author Response

Dear Expert,

Thank you very much for your valuable suggestions. The related content has been commented in the attachment. Please check it when you have a moment.

Reviewer 2 Report

Comments and Suggestions for Authors

The authors made a huge work. Nevertheless, I have two objections:

1. Explanation of the Figure 2 must be expanded. It it to short, but it seems to be very important

2. Equations (Line 496, 518, 537): I suggest to write them in a form of a fraction. They will be easier to understand

At the end, I would like to suggest the manuscript to be accepted after minor revison.

Author Response

(The authors gave the same response as above.)

Reviewer 3 Report

Comments and Suggestions for Authors

Dear Authors,

In my opinion the submitted manuscript titled „Unraveling the Genetic Control of Pigment Accumulation in Physalis Fruits” is generally well-written. Moreover, due to interesting topic and valuable results it might interest an international audience. However I have found some imperfection, which- in my opinion- should be corrected or at east clarified before an eventual publication. Please, find them in below listed points:

1.       Intoduction chapter. In my opinion the object of investigations Physalis pubescens and Physalis alkekengi should be more detailedly described. Please, focus on natural range, habitat affiliation, lifespan, life form  and particularly properties and use of fruits. I suggest to add information about the current state of knowledge on genetic control of pigment in Physalis taxa and point out the gap, that will justify the aims Your investigations. Lines 76-88- instead of this text I suggest to list in points the specific aims of investigations.

2.       In current form Figures 2,4,5 and 6 are hardly legible. Their quality should be improved.

3.       Discussion chapter. I encourage Authors to compare obtained results with publications referring to other species from Physalis genus (i fit is possibile).

4.       Material and methods chapter. Lines 618-624 in my opinion the performer statistical analyses should be described more detailedly.

5.       Please, look into following papers. Perhaps, some of them will be useful in manuscript corrections:

·         Wen, X., Heller, A., Wang, K. et al. Carotenogenesis and chromoplast development during ripening of yellow, orange and red colored Physalis fruit. Planta 251, 95 (2020). https://doi.org/10.1007/s00425-020-03383-5

·         Sheikha et al. 2008. Physico-chemical Properties and Biochemical Composition of Physalis (Physalis pubescens L.) Fruits. Food 2 (2), 124-130.

·         Wang L, Li J, Zhao J and He C (2015) Evolutionary developmental genetics of fruit morphological variation within the Solanaceae. Front. Plant Sci. 6:248. https://doi.org/10.3389/fpls.2015.00248

·         Patel PR, Gol NB, Ramana Rao TV. Physiochemical changes in sunberry (Physalis minima L.) fruit during growth and ripening. Fruits. 2011;66(1):37-46. https://doi.org/10.1051/fruits/2010039

·         Gonzali, S.; Perata, P. Fruit Colour and Novel Mechanisms of Genetic Regulation of Pigment Production in Tomato Fruits. Horticulturae 2021, 7, 259. https://doi.org/10.3390/horticulturae7080259

Author Response

Dear Expert,

Thank you for your assistance. I have greatly benefited from reading the literature you recommended. Additionally, I appreciate the valuable suggestions you have offered. The relevant content has been noted in the attachment. Please check it when you have a moment.

Round 2

Reviewer 1 Report

Comments and Suggestions for Authors

The authors have revised the manuscript in accordance with the comments, and there are no further comments.

Author Response

Dear Expert,

Thank you for your valuable suggestions on the manuscript. We also appreciate your recognition of our work.

Reviewer 3 Report

Comments and Suggestions for Authors

Dear Authors,

Thank You very much for implementing my suggestions. In my opinion manuscript is sufficiently improved. My only concern refers to legibility of Figure 5 and 6a due to small font size.

Author Response

Dear Expert, Thank you for your valuable feedback. Figures 5 and 6a have been revised as suggested. We appreciate your precious comments amid your busy schedule and are grateful for your support.